# PokéChamp: an Expert-level Minimax Language Agent for Competitive Pokémon

## Abstract

We introduce `PokéChamp`, a Large Language Model (LLM) powered game-theoretic aware agent for two-player competitive Pokémon battles, that uses an LLM prior and collected high-Elo human data to model minimax search without any additional neural network training. `PokéChamp` uses a depth-limited minimax search online where the LLM replaces three key components: 1) action sampling from the LLM guided by prompts (including from a damage calculation tool), 2) opponent-modeling via the historical likelihood of actions from our dataset to model the effect of LLM-predicted opponent actions, and 3) state value calculation for the LLM to reflect on each intrinsic state. `PokéChamp` outperforms all existing LLM-based (76%) and rule-based bots (84%) by an enormous margin, including winning consistently (64%) against prior human-parity work run with a frontier model, GPT 4-o, while using an open-source 8 billion parameter Llama 3.1 model. `PokéChamp` achieves expert performance in the top 10% of players on the online ladder against competitive human players at an Elo of 1500. Finally, we collect the largest Pokémon battling dataset, including 1 million+ games with 150k+ high Elo games, prepare a series of battling benchmarks based on real player data and puzzles to analyze specific battling abilities, and provide crucial updates to the local game engine. Our code is available online.

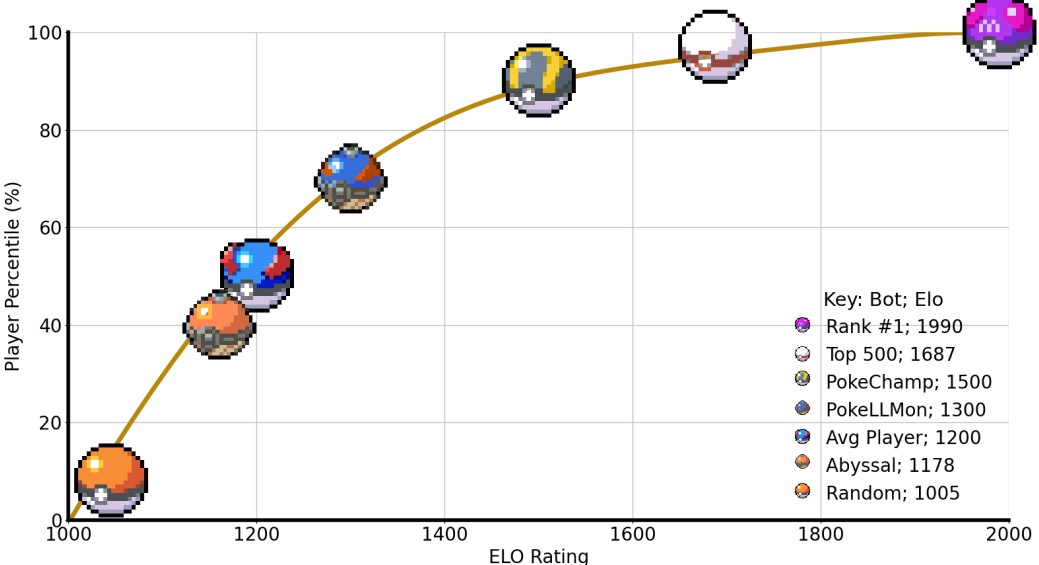

Figure 1: `PokéChamp` achieves the 90% percentile of players and a 1500 Elo rating against real players. Higher Elo and percentile denote better performance.

# 1 INTRODUCTION

A desirable element of a language agent is to be able to approach Nash solutions for competitive tasks. Previous work in reinforcement learning Campbell et al. (2002); Silver et al. (2017; 2016); Brown & Sandholm (2019; 2018); Vinyals et al. (2019); Berner et al. (2019) finds the best performance through *tabula rasa* self-play or train large language models via imitation learning and human-regularized self-play (FAIR). Language agents have the ability to leverage prior information about background and task strategies and apply it to new situations. Recent advances in these text agents have shown that they lack basic planning capabilities Topsakal & Harper (2024), perform worse than hardcoded heuristic bots for games Küttler et al. (2020), and have issues understanding the basic game mechanics Hu et al. (2024b).

Let us consider competitive Pokémon as an example. With a vast number of Pokémon species that each have their own unique abilities, moves, typing, special mechanics, and range of stats, the total number of possible states is on the order of $10^{354}$ The-Third-Build (2022) and this is only for turn 1. Information about the opponent's team is only partially observable, which keeps the search space from shrinking once the opponent's team is revealed. Pokémon battles can last anywhere from 6 to over 100 turns, meaning exhaustive self-play can be quite computationally intractable. We argue that an informative prior can help limit minimax search to the space of human strategies. Large language models (LLMs) have the potential to be trained on text datasets that include information about Pokémon, but are not exhaustive since they are not the focus of training. Thus, we want an agent that can harness the prior of large language models to: (1) **propose optimal actions** to provide highly likely, diverse actions for potential strategies, (2) **accurately model the opponent** based on their move history, team, and strategy based on their skill-level, and (3) **reflect** on an internally planned game trajectory without requiring a terminal win/lose state.

Thus, to achieve these mechanisms, we introduce `PokéChamp`, a competitive language agent that achieves human-expert level performance in the two-player turn-based battles in *Pokémon*. `PokéChamp` uses a large language model to power generative minimax tree search through the following elements: (1) action branching from **tool-use and LLM sampling**; (2) **LLM-based opponent-modeling** via information from our **collected historical data** of real Pokémon battles to provide likely team compositions, which is prompted to the LLM to provide opponent actions; and (3) using a score rating guide prompt and the internally predicted state from our world model engine, provide an **LLM-generated value calculation** of leaf nodes. Based on the depth of our tree, the sum of the values of the leaf nodes is backpropagated up through the tree to output the most likely action. The LLM acts as a black box, in which it can be switched out based on one's compute availability and used with better frontier LLMs as they are developed. Our approach prevents the need for additional training or fine-tuning on Pokémon-specific data.

In order to successfully plan $k$ steps into the future battle, our agent needs a proper world model. Due to partial observability, we cannot just use the game engine as the world model. Rather, we developed our own local game engine in order to adequately address the intrinsic planning capabilities. We developed a tool, which we simply call the damage calculator (dmg calc), that mathematically calculates the core game engine capabilities in combination with loading historical data from real player games in order to load likely stats for the opponent's team. The historical data comes from our Pokémon battling dataset, which, as of this writing, contains over 1 million games from various Elos and game modes.

We use our dataset, which is the first and largest Pokémon battling dataset, to establish a set of benchmark puzzles to understand key mechanics and strategies commonly found in battles. We establish a move prediction and opponent modeling benchmark based on high-elo human data to show the limitations of prompting LLMs. Our 1v1 benchmark provides an opportunity to evaluate the competency of our bot to choose the best moves for individual matchups.

Empirically, `PokéChamp` demonstrates expert-level performance in Pokémon battles. Our agent is able to choose optimal actions with a short minimax planning lookahead. We evaluate `PokéChamp` in an arena setup against other competitive Pokémon bots, including heuristic bots and an LLM-based agent PokéLLMon Hu et al. (2024b), in two highly played game modes: Generation 8 Random Battles (gen8randombattles) and Generation 9 OverUsed Meta (gen9ou). *PokéChamp* outperforms all other bots and AI agents, with the largest margin being in gen9ou with a 76% winrate against the strongest LLM-based bot and an 84% winrate against the strongest heuristic bot. Our method

is also able to bootstrap smaller language models such as Llama3.1:8b to win consistently (64%) against prior LLM-based bots with GPT-4o. In online ladder battles, we find that our language agent, `PokéChamp`, achieves an expert rating on 1500 Elo, which is just shy of the top 500 players.

## 2 MATHEMATICAL FORMALIZATION

**Partially observable Markov games:**    We consider finite-horizon, two-player zero-sum Markov Games with partial observability, which can be described as a tuple $\text{POMG}(H, \mathcal{S}, \mathcal{X}, \mathcal{Y}, \mathcal{A}, \mathcal{B}, \mathbb{P}, r)$, where

- $H$ is the horizon length;
- $\mathcal{S} = \bigcup_{h \in [H]} \mathcal{S}_h$ is the (underlying) state space with $|\mathcal{S}_h| = S_h$ and $\sum_{h=1}^{H} S_h = S$;
- $\mathcal{X} = \bigcup_{h \in [H]} \mathcal{X}_h$ is the space of information sets (henceforth *infosets*) for the *max-player* with $|\mathcal{X}_h| = X_h$ and $X = \sum_{h=1}^{H} X_h$. At any state $s_h \in \mathcal{S}_h$, the max-player only observes the infoset $x_h = x(s_h) \in \mathcal{X}_h$, where $x : \mathcal{S} \to \mathcal{X}$ is the emission function for the max-player;
- $\mathcal{Y} = \bigcup_{h \in [H]} \mathcal{Y}_h$ is the space of infosets for the *min-player* with $|\mathcal{Y}_h| = Y_h$ and $Y = \sum_{h=1}^{H} Y_h$. An infoset $y_h$ and the emission function $y : \mathcal{S} \to \mathcal{Y}$ are defined similarly.
- $\mathcal{A}, \mathcal{B}$ are the action spaces for the max-player and min-player, respectively, with $|\mathcal{A}| = A$ and $|\mathcal{B}| = B$;
- $\mathbb{P} = \{p_0(\cdot) \in \Delta(\mathcal{S}_1)\} \cup \{p_h(\cdot|s_h, a_h, b_h) \in \Delta(\mathcal{S}_{h+1})\}_{(s_h, a_h, b_h) \in \mathcal{S}_h \times \mathcal{A} \times \mathcal{B}, \, h \in [H-1]}$ are the transition probabilities, where $p_1(s_1)$ is the probability of the initial state being $s_1$, and $p_h(s_{h+1}|s_h, a_h, b_h)$ is the probability of transitting to $s_{h+1}$ given state-action $(s_h, a_h, b_h)$ at step $h$;
- $r = \{r_h(s_h, a_h, b_h) \in [0, 1]\}_{(s_h, a_h, b_h) \in \mathcal{S}_h \times \mathcal{A} \times \mathcal{B}}$ is the binary reward for winning a game.

**Policies, value functions:**    As we consider partially observability, each player's policy can only depend on the infoset rather than the underlying state. A policy for the max-player is denoted by $\mu = \{\mu_h(\cdot|x_h) \in \Delta(\mathcal{A})\}_{h \in [H], x_h \in \mathcal{X}_h}$, where $\mu_h(a_h|x_h)$ is the probability of taking action $a_h \in \mathcal{A}$ at infoset $x_h \in \mathcal{X}_h$. Similarly, a policy for the min-player is denoted by $\nu = \{\nu_h(\cdot|y_h) \in \Delta(\mathcal{B})\}_{h \in [H], y_h \in \mathcal{Y}_h}$. A trajectory for the max player takes the form $(x_1, a_1, r_1, x_2, \ldots, x_H, a_H, r_H)$, where $a_h \sim \mu_h(\cdot|x_h)$, and the rewards and infoset transitions depend on the (unseen) opponent's actions and underlying state transition. The overall game value for any (product) policy $(\mu, \nu)$ is denoted by $V^{\mu,\nu} = \mathbb{E}_{\mu,\nu}[\sum_{h=1}^{H} r_h(s_h, a_h, b_h)]$. The max-player aims to maximize the value, whereas the min-player aims to minimize the value.

**Tree structure and perfect recall:**    We use a POMG with tree structure and the perfect recall assumption as our formulation for IIEFGs, following (Kozuno et al., 2021). We assume that our POMG has a *tree structure*: For any $h$ and $s_h \in \mathcal{S}_h$, there exists a unique history $(s_1, a_1, b_1, \ldots, s_{h-1}, a_{h-1}, b_{h-1})$ of past states and actions that leads to $s_h$. We also assume that both players have *perfect recall*: For any $h$ and any infoset $x_h \in \mathcal{X}_h$ for the max-player, there exists a unique history $(x_1, a_1, \ldots, x_{h-1}, a_{h-1})$ of past infosets and max-player actions that leads to $x_h$ (and similarly for the min-player).

**Best response and Nash equilibrium:**    For a two-player game, for a max-player's policy $\mu$, there exists a best response policy $\nu^{\dagger}(\mu)$ by the min player, which satisfies $V^{\mu,\nu^{\dagger}(\mu)}(s) = \inf_{\nu} V^{\mu,\nu}(s)$ for any $s \in \mathcal{S}$ and similarly for the min-player's best response $\nu$ for $\mu^{\dagger}(\nu)$ satisfying $V^{\dagger,\nu} = \sup_{\mu} V^{\mu,\nu}$.

**Definition:** (Nash equilibrium). *The Nash equilibrium is defined as a pair of policies $(\mu^*, \nu^*)$ that provide the optimal action choice when the opponent's chooses the best response, which implies,* $V^{\mu^*,\dagger}(s) = \sup_{\mu} V^{\mu,\dagger}(s), \; V^{\dagger,\nu}(s) = \inf_{\nu} V^{\mu,\nu}(s), \text{for all } s \in \mathcal{S}.$

Nash equilibrium strategies satisfy the minimax equation, given by,

$$\sup_{\mu} \inf_{\nu} V^{\mu,\nu}(s) = V^{\mu^*,\nu^*}(s) = \inf_{\nu} \sup_{\mu} V^{\mu,\nu}(s). \tag{1}$$

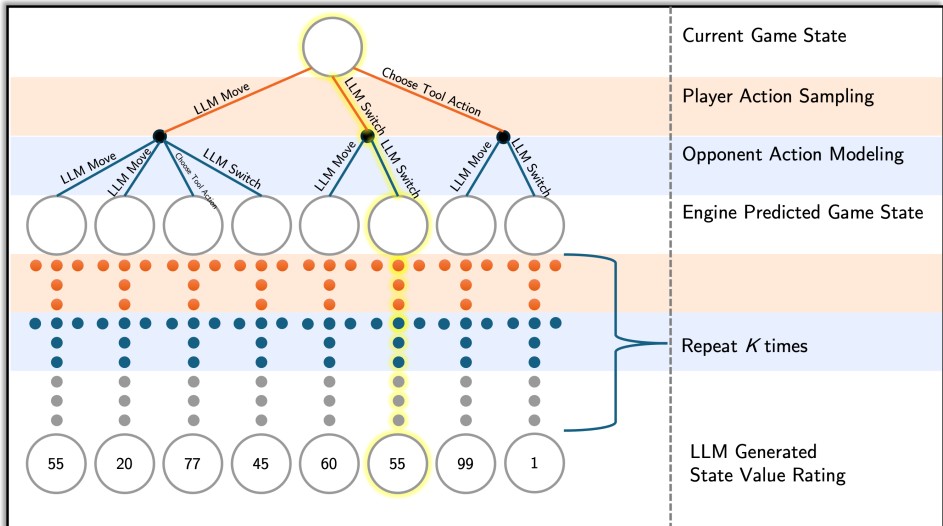

Figure 2: PokéChamp replaces three components of minimax tree search with LLM-based generations: (1) sampling potential actions for the player corresponding to the first part of the edge between states., (2) modeling the opponent and sampling opponent actions corresponding to the second part of the edge between states, and (3) generating a potential game state value based on the depth $K$ cutoff. PokéChamp provides the action with the best minimax value to be used in battle.

## 3 POKÉCHAMP

PokéChamp consists of three novel components that perform a minimax tree search: (1) policy sampling via a combination of LLM and tool-use, (2) opponent modeling via historical data and LLM generation, and (3) an LLM-generated value function. To output the best action, the value is recursed through the tree to choose the action with the maximum value for player moves and the minimum value for opponent moves. Full prompts are available in the Appendix. See figure 2.

### 3.1 ACTION SAMPLING

Using the input prompt, PokéChamp generates an edge for both action types: move and switch. The top move choice by the damage calculator are top switch choice from the Abyssal bot, which rates matchups based on a fixed set of rules, is also used as a branch of the search tree. The edges of the minimax search tree are made of action samples from the LLM. The input prompt consists of the following features:

- **LLM Team strategy:** The LLM is asked to generate an overall team strategy based on the summary of the player's team and the opponent's team;
- **State:** This includes the available team, items, opponent's visible team, etc;
- **Battle history:** This includes the information from the last $N$ turns.;
- **Damage calculation:** For each of the current Pokémon's damaging and stat raising moves, the number of turns to KO the opponent's Pokémon is calculated. Additionally, the number of turns including switching to another Pokémon and then KO'ing the opponent's current Pokémon is calculated. This can be thought of as a depth-first search heuristic with a cutoff at KO;
- **Available actions**: The current pokémon has up to 4 available moves, up to 5 available switch options, and a special mechanic such as dynamax or terastallize, if available.

**Damage Calculator** The damage calculator provides a feature for individual Pokémon matchups. For each matchup, the damage calculator predicts how many times it takes each damaging move to KO the opponent's Pokémon. If the Pokémon has a status move, we advance a local simulation of

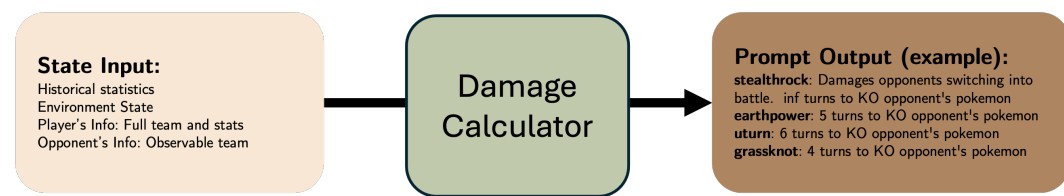

Figure 3: `PokéChamp` uses damage calculator prompts to eliminate the need for the LLM to perform exact mathematical computations.

the world to update the state and then try each move with the damage calculator. This solves an abridged tree search to output the best move. We create our own damage calculator that uses the following equation Bulbapedia (2024),

$$\text{Damage} = \left( \frac{1}{50} \left( \left( \frac{2}{5} \cdot \text{Level} + 2 \right) \cdot \text{Power} \cdot \frac{A}{D} \right) + 2 \right) \cdot \text{Other Mechanics} \tag{2}$$

$A$ is the attack or special attack stat. $D$ is the defense or special defense stat. These stats our known for our own team. The other items are special stats and mechanics that need to be tracked by our clientside game engine. Please see the appendix for the list of all other mechanics calculated in equation 3 and figure 3 for an example damage calculator prompt output. A full prompt output is available in the appendix in listing 1.

## 3.2 Opponent Modeling

Unlike with the agent's team, the opponent's $A$ attack and $D$ defense is unknown. Using historical data, we can estimate the likelihood of each set of stats the opponent may choose for their EV/IVs. We can either choose the most likely option or sample from the set with probability equal to their likelihood. The prompt for generating the likely opponent actions from the LLM is similar to the action sampling. Afterwards, we can use our proprietary damage calculator to predict next turn state given the edges from the player action and opponent action. Perfect prediction is not possible to be perfect due to stochastic transition function that includes the `random` element of move damage and `accuracy` and moves. We choose to be optimistic and choose the best option assuming maximum damage, which means the high end of the random spectrum and 100% accuracy.

## 3.3 Value Function

Live games provide each player with 150 seconds for the entire match. An additional up to 15 seconds is added to each player's clock after their turn. Due to time limitations, we are unable to perform exhaustive minimax tree search across all player and opponent action possibilities. Thus, we must perform a value calculation at leaf nodes to estimate the utility of a state. We ask the LLM to provide a score for the internal state based on the following prompt features:

- **Add points:** based on the effectiveness of current moves, number of player's Pokémon remaining, and overall player's likelihood to win.

- **Subtract points:** based on excessive switching, effectiveness of opponent's current moves, faster opponent speed, number of opponent's remaining Pokémon, and the strength of the remaining Pokémon.

After the value is generated by the LLM, the value is propagated up the tree to the root, which chooses the player actions with the highest score and the opponent actions that correspond to the lowest scores , which satisfy equation 1.

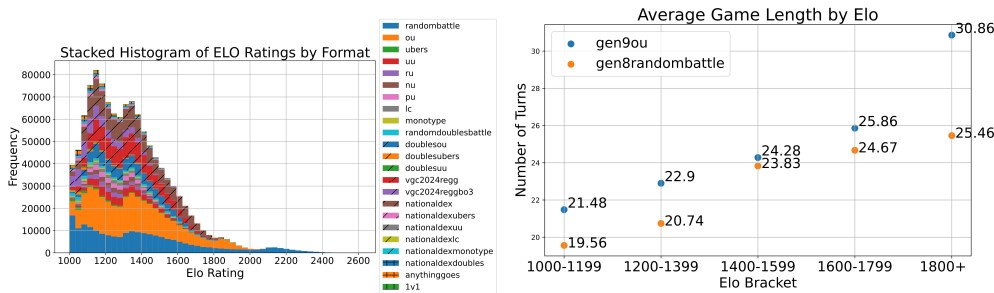

Figure 4: **Left:** Elo distribution for collected battles for each gamemode. **Right:** Game length by elo (scatter plot).

Table 1: Player action prediction accuracy by Elo. Random performance is 7% for player prediction and less than 1% due for opponent prediction to partial observability.

| Model | Elo | Top 1 | Top 2 | Top 3 | Top 4 | Top 5 |
|---|---|---|---|---|---|---|
| Player | 1200 | 30% | 40% | 48% | 53% | 58% |
| Player | 1400 | 26% | 23% | 30% | 32% | 43% |
| Player | 1600 | 27% | 30% | 39% | 44% | 53% |
| Player | 1800 | 30% | 42% | 55% | 62% | 66% |
| Opponent | 1200 | 16% | 30% | 40% | 46% | 53% |
| Opponent | 1400 | 16% | 17% | 20% | 26% | 39% |
| Opponent | 1600 | 13% | 21% | 26% | 35% | 40% |
| Opponent | 1800 | 15% | 29% | 40% | 50% | 53% |

## 4 DATASET AND PUZZLES

### 4.1 POKÉMON BATTLING DATASET

We scraped over 1 million games of replay data from Pokémon Showdown for use in our opponent prediction model. Over 150k replays are from high Elo games (>1600 Elo). An analysis of the data can be seen in figure 4. The data shows the distribution of Elos follows a multi-modal distribution with 2 modes, approximately 1150 and 1350. It shows that higher Elo matches typically take longer. More information about the gamemodes are available in the appendix B.

### 4.2 HUMAN ACTION PREDICTION

The replay data provides information about battles from the perspective of the opponent. This means that not all stats are available. Thus, in order to perform prediction, we must first reverse engineer the moves, team, and stats from the replay data. Afterwards, we can fill in additional switching and move options from what is historically likely. Feeding this information into our prompt generator will give our bot the ability to predict player and opponent actions.

We compare the accuracy of these actions with the historical data across various Elos. Our results are shown in table 3.3. The player prediction accuracy for PokéChamp varies between 26% and 30% as Elo increases while the opponent prediction accuracy is lower, between 13% and 16%. This shows that predicting opponent actions is more difficult given our state information. The opponent prediction performs a little better than random, while the player prediction performs much better than random. However, there is still quite an improvement to be made. There may also be more than one *correct* action possible. Many strategies may be equally optimal option. Rather, the performance may simply have to do with player preference.

### 4.3 BENCHMARK PUZZLES

We set up a series of benchmark puzzles to determine whether the agent is able to correctly approach each of the battle mechanics: choosing the best move in individual matchups and using special

Figure 5: PokeChamp in Figure 8 (right) understands the changed weakness of Roaring Moon and decides to switch after Glimmora is chosen, as terastallizing has caused Roaring Moon to become weak to rock-types. PokeChamp in Figure 8 (left) uses dynamax to increase its hit points and cause its moves to deal more damage, allowing it to knock out two Pokémon in a row.

Table 2: 1v1 benchmark performance comparison.

| Model | Win Rate (%) |
|---|---|
| Pokellmon | 76% |
| **Pokéchamp (Ours)** | **86%** |

mechanics such as terastallization and dynamaxing to one's advantage. More puzzles and analysis of situations are available in the appendix D.2.

### 4.3.1 ONE VERSUS ONE

In order to understand if our agent is able to choose the the correct sequence of moves to KO the opponent's Pokémon before they can KO the player's Pokémon, we create a 1v1 benchmark. We select matchups from the gen8randombattles meta. Each matchup consists of one Pokémon on each team. In order to ensure that there is a feasible win condition, we reject samples that are not able to be won by the Abyssal bot. Though, we note due to the stochasticity of move damage, this does not ensure 100% is possible every time. We sample 1000 1v1 configurations and report the performance in table 4.3.2. Our method is able to win 10% more consistently due to the use of the damage calculator in the lookahead.

### 4.3.2 SPECIAL MECHANICS: TERASTALLIZATION AND DYNAMAX

Our world model and prompting mechanism for PokeChamp allow it to understand and use generation-specific game mechanics, in this case Dynamax and Terastallization. It is informed of what the mechanics do, and the damage calculator is used to inform PokeChamp about the different outcomes if it were to use the mechanics.

## 5 EVALUATION

In this section, we compare `PokéChamp` with baseline algorithms in the Gen 9 OU format. Then we evaluate experiments on the online ladder. We provide additional experiments in the appendix E for the Gen 8 Random Battles format.

### 5.1 BASELINES

We compare with the best open source bots and a SOTA LLM-based agent. We perform each experiment with at least 25 matches between any two individual methods per experiment. Thus, each method is run in at least 100 games when computing the Elo. The LLM agents use either Llama3.1:8b Dubey et al. (2024) or GPT-4o-2024-05-13 Achiam et al. (2023).

**PokéLLMon** Hu et al. (2024b) is a prompting-based language agent that provide state features and a battle history to the LLM to produce an action. It uses the self-consistency Wang et al. (2022) prompting method to output the most likely action.

Table 3: Battling in Gen 9 OU without the terastallize mechanic in mirror matchups.

| Bot Method | Language Model | Win Rate vs. Abyssal (%) |
|---|---|---|
| **PokéChamp (Ours)** | **GPT-4o** | **90%** |
| PokéChamp (Ours) | Llama 3.1:8b | 83% |
| PokéLLMon | GPT-4o | 60% |
| Dmg Calc | N/A | 56% |

Table 4: Battling in Gen 9 OU with the terastallize mechanic and custom teams.

| Bot Method | Language Model | Win Rate vs. Abyssal (%) | Elo | Avg # Turns |
|---|---|---|---|---|
| **PokéChamp (Ours)** | **GPT-4o** | **84%** | **1268** | **15.7** |
| PokéChamp (Ours) | Llama 3.1:8b | 56% | 1204 | 16.9 |
| PokéLLMon | GPT-4o | 40% | 1020 | 22.6 |
| Abyssal | N/A | N/A | 1117 | 17.9 |
| Dmg Calc | N/A | 44% | 1107 | 17.9 |
| Max Power | N/A | 16% | 885 | 19.5 |
| Random | N/A | 0% | 399 | 21.2 |

**Abyssal Bot** is a heuristic bot used as the most challenging game intelligence in the Pokémon video games. The bot has rules to setup stat-boosting moves and select the highest damage actions, taking into consideration typing advantages and other mechanics. It also rates matchups so that they try to switch to the best matchups to defeat the opponent.

**Dmg Calc Bot** is a planning-based bot that we created to output the action that provides the greatest damage to the current enemy Pokémon based on our proprietary game engine.

**Max Power Bot** selects the move with the highest power level regardless of the typing advantage or disadvantage.

**Random** selects randomly generated actions.

## 5.2 FIXED BATTLE AND META

Our main experiments focus on using a fixed battle, which means teams are decided by the players in advance, and meta, which is the allowed Pokémon and special game mechanics. We choose the most popular fixed battle meta, the Gen 9 OU (OverUsed) format with the special terastallizing mechanic, which allows Pokémon to change their typing. The tier is based on usage statistics and changes over time. Pokémon that are too powerful for OU are banned to Ubers, and those rarely used are placed in lower tiers.

In table 5.1, we present the results of the Gen 9 OU battles. Each battle, the player and opponent battle with one of five available Pokémon Showdown-approved teams. `PokéChamp`-GPT performs the strongest out of any method with an 84% winrate over the Abyssal bot and an overall Elo of 1268. Notably, `PokéChamp`-Llama outperforms PokéLLMon, which is powered by GPT-4o. This shows that even an 8 billion parameter model can outperform a frontier model if provided with planning algorithms and tools. While the performance against the Abyssal bot looks increasing with the complexity of the method, individual matchups show interesting behavior. In the left figure 6, we present the individual winrates between any two methods. For instance, the `PokéChamp`-GPT has a higher winrate over the Abyssal bot than the Llama version, but the Llama version has a higher winrate over PokéLLMon. The average number of turns correlates slightly with the overall winrate against the Abyssal bot and performance of the bot.

In order to analyze the effect of team composition on the winrate of the method, we analyze the In the right figure 6, we see that the selected team can change the winrate up to 8%. In our custom team setup, we ensure the same number of each matchup to ensure balance. Additionally, the terrastallize mechanic may have a strong effect against heuristic bot performance since they do not have rules setup to invoke them. In order to further study these effects, we explore battles with mirror matchups without the special terastallize mechanic. Mirror matchup simply means that both the player and the opponent have the same team. In table 5.1, we analyze the results of Gen 9 OU battles with a mirror

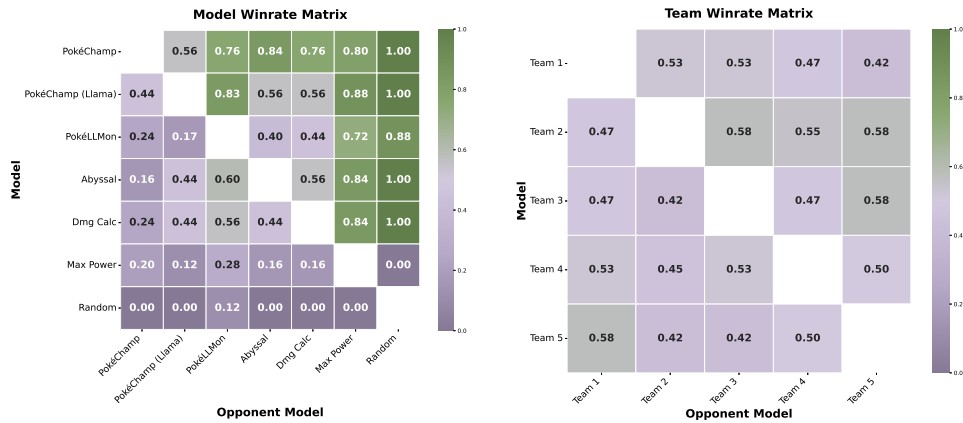

Figure 6: **Left:** Gen 9 OU matchup matrix per method. **Right:** Gen 9 OU matchup matrix per team

Table 5: `PokéChamp` versus online ladder players.

| Winrate (%) | Time Loss Rate (%) | Elo | Total Games |
|---|---|---|---|
| 76% | 33% | 1500 | 50 |

matchup team composition. The agents perform better against the heuristic bot on average. This implies that the team composition has a large role on the performance. Additionally, the terastallize effect has a less noticeable change on the winrate than the effect of the composition. `PokéChamp` achieves a 90% winrate over the Abyssal bot in this mode.

## 5.3 ONLINE LADDER

Competitive Pokémon is available to play online at Pokémon Showdown, where 10,000+ players and over 2000 games are active at any given time. Each battle has a time limit of 2 minutes and 30 seconds with an additional 15 seconds gained each turn. Each season, players have their Elo reset to 1000, the default rating, ensuring that each player's Elo corresponds to an accurate skill level.

`PokéChamp` battles real opponents on the ladder for a total of 50 games in the Gen 9 OU meta. The fastest turns of our method take under 10 seconds to run. However, when there are full actions our available for the player and opponent, our method can take as long as one minute (depending on the speed of the LLM calls). Thus, 33% of the total games were lost due to running out of time on the clock. In table 5, we show the results of the battles. Of the finished games, `PokéChamp` achieves a 76% winrate against real human opponents. With the opponent's average rating of 1300, we have an estimated Elo rating of 1500, which is in the top 10% of all players.

`PokéChamp` performs particularly poorly against two key strategies: stall and excessive switching. We provide additional discussion regarding these strategies in the appendix and in figure 7. See more details in sections D.1 and D.2. This is most likely due to the limited lookahead in order to satisfy clock constraints and the accuracy of opponent modeling in-context.

## 6 RELATED WORK

### 6.1 COMPETITIVE GAMES

Previous work in competitive games, such as Chess Campbell et al. (2002); Silver et al. (2017), Go Silver et al. (2016), poker Brown & Sandholm (2019; 2018), Starcraft II Vinyals et al. (2019), and Dota 2 Berner et al. (2019), uses reinforcement learning to achieve superhuman performance by training exhaustively *tabula rasa*. Other prior efforts solve competitive fighting games such as Streetfighter Li et al. (2024) through multi-agent reinforcement learning, simulating a ladder-style tournament where AI agents compete in one-on-one matches. This trend has continued to find human-level performance in more realistic game simulations such as Gran Turismo Sophy Wurman et al. (2022), which outperformed world-class human drivers in the racing game Gran Turismo

Sport. Some efforts in this direction have been tried in Pokémon battles Huang & Lee (2019), demonstrating competitive performance against heuristic bots, as well as human-level effort in the video game series Whidden (2023) through iterative learning and reward optimization. Unlike prior work, `PokéChamp` does not train exhaustively Pokémon. In fact, our method does not explicitly train at all, yet performs at an expert level.

## 6.2 Language Agents for Games

There is growing interest in language agents for games Hu et al. (2024a), but they still fundamentally fail at basic planning algorithms. LLMs are still unable to play the Nash strategy for Tic-Tac-Toe Topsakal & Harper (2024), which is a simple $3 \times 3$ solved game. This shows the need for additional planning augmentation for LLMs. Nethack Küttler et al. (2020) is a roguelike game designed to test open-ended and long context reinforcement learning agents. However, an LLM-powered Nethack agent Jeurissen et al. (2024) still performs poorly compared to an extensive heuristic bot. Other work on LLMs for games uses prompting algorithms to overcome planning limitations of LLMs for Pokémon Hu et al. (2024b), StarCraft Ma et al. (2023), and mixed cooperative-competitive games such as Avalon Shi et al. (2023); Stepputtis et al. (2023). In another direction, open-world games are being explored by LLMs such as Voyager Wang et al. (2023) learning skills in Minecraft or Spring Wu et al. (2024), which understands how to play a game by reading the strategy guide. LLMs have also been finetuned specifically based on human game data in Diplomacy (FAIR). Using finetuning and reinforcement learning with LLMs has seen interest in model distillation Nalty & Rosenthal, two-player online reinforcement learning for finetuning LLMs Zhou et al. (2024), and using an LLM to generate reward feedback for a reinforcement learning agent Klissarov et al. (2023).

## 6.3 Prompting and Planning

Recent advancements in prompting techniques and planning algorithms have significantly enhanced the reasoning capabilities of large language models. Chain-of-thought prompting Wei et al. (2022) improves models' reasoning by providing step-by-step examples, while self-consistency Wang et al. (2022), boosts performance by sampling multiple reasoning paths. The Tree of Thoughts Yao et al. (2024) framework generalizes this approach, enabling models to explore and evaluate multiple reasoning paths. In a similar vein, the ReAct Yao et al. (2022) framework interleaves reasoning traces with text actions, enhancing performance on tasks requiring both reasoning and interaction. Other work has demonstrated that language models can serve as both a world model and reasoning agent in their Reasoning via Planning (RAP) approach Hao et al. (2023). This concept is further explored in work Zhao et al. (2023) which leverages language models as commonsense knowledge sources for large-scale task planning. The integration of search algorithms with language models has also shown promise, as evidenced by the TS-LLM Feng et al. (2023) framework, which applies AlphaZero-like tree search to guide model decoding and training. Additionally, researchers have explored reinforcement learning techniques to improve models' self-correction abilities, as seen in the SCoRe Kumar et al. (2024) method and the Reflexion Shinn et al. (2024) framework, which uses linguistic feedback and episodic memory to enhance performance without extensive fine-tuning.

## 7 Conclusion

In this paper, we introduce `PokéChamp`, which augments minimax tree search with the following LLM-based components: (1) action sampling, (2) opponent modeling, and (3) a state value function. `PokéChamp` achieves state-of-the-art performance against heuristic and LLM-based bots and expert performance against real players on the online ladder. Further performance enhancements are currently limited by the accuracy of opponent modeling and the method's online computational budget. By increasing the breadth and depth size of the search, we expect to see further improvement's to the performance of the method. Additionally, our work can be taken advantage of adversarially due to static opponent modeling. Our work leaves open challenges in opponent modeling and generative minimax planning for future work exploring competitive multiagent settings and *future superhuman performance* in Pokémon battling.

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

# A  APPENDIX

# B  BACKGROUND

## B.1  GAME MECHANICS

Extensive details of the game mechanics can be found here: `https://pokemonshowdown.com/intro`. From Pokémon generation 6 and forward, special generation-exclusive mechanics were introduced that add another layer of dynamics to Pokémon Battles. In generations 8 and 9, the two generations used for testing with the latter being the newest generation, the special mechanics are 'Dynamaxing' and 'Terastallization' respectively. 'Dynamaxing' a Pokémon increases its health as well as the damage of its moves for 3 turns. 'Terastallizing' a Pokémon changes its typing which can alter its strengths and weaknesses for the rest of the battle's duration. Our world model expands on the world model used in PokeLLMon to allow the battling agent to make more accurate opponent predictions, and allow it the ability to 'dynamax' or 'terastallize' itself, better mimicking a human player.

### B.1.1  TEAM BATTLES

The Smogon singles formats are Ubers, OU (Overused), UU (Underused), RU (Rarely-used), NU (Neverused), LC (Little Cup). These formats are all singles bring-6-choose-6. They're named after their corresponding Smogon tier (except LC).

### B.1.2  RANDOM BATTLES

Random battles are a specific type of team battles that use the OU set of Pokémon with randomly procedurally generated team of six Pokémon and movesets. Due to the procedural rule generation, random battles are not completely random Mitarai & Pujo (2021). See `https://hypixel.net/threads/random-battles-the-worst-meta-game-in-pokemon.5112201/`.

## B.2  PARTIAL OBSERVABILITY

The game of Pokémon is partially observable. The opponent's moves, abilities, and sometimes even Pokémon on their team are not viewable until after use.

## B.3  ELO AND ONLINE EVALUATION

Pokémon Showdown provides an online skill-based matchmaking ladder based on the Elo system Contributors to Wikimedia projects (2024).

## B.4  POKÉMON AI

There have been multiple attempts to create an AI Pokémon battling agent. However, most fail due to a large branching factor, which is estimated to be at least over 54000 when using a coarse-grain discretization for damage randomness approximation and limiting the number of eligible Pokémon to 100.

The Pokémon battling AI built into the original Pokémon games follows a set of heuristic rules. The best being the Abyssal bot, which has an 1178 Elo rating in the OU gamemode ladder on Pokémon Showdown. The ladder has a minimum Elo of 1000.

There are also damage calculators such as showdex `https://github.com/doshidak/showdex`, the Smogon damage calculator `https://github.com/smogon/damage-calc`, and the GUI version: `https://calc.Pokémonshowdown.com/`, that calculate the likelihood range of damage and predict the corresponding number of moves required to KO a Pokémon. This calculates the EVs and IVs in random battles based on the data available.

The FSAI The-Third-Build (2022) move predictor uses machine learning to predict the probability of what opponent move is used next. Then it uses an inverse damage calculator and combines this with a small expectiminimax calculation to choose the best move. The method has the following

failure cases: FSAI is unable to accurately predict the last few moves at the end of a battle due to long sequences not found in its data. After turn 14, FSAI is unpredictable, creating a "choking" problem. This method does not exceed 1600 Elo on gamemodes with fixed teams (not random teams). Unfortunately, this project is deprecable and unable to be directly compared with due to a lack of an open-source implementation or explicit methodology.

## C  DAMAGE CALCULATION

The full damage calculation is given by the following equation,

$$
\begin{aligned}
\text{Damage} = \bigg( \frac{1}{50} \bigg( \bigg( \frac{2}{5} \cdot \text{Level} + 2 \bigg) \cdot \text{Power} \cdot \frac{A}{D} \bigg) + 2 \bigg) \\
\cdot \text{Targets} \\
\cdot \text{PB} \\
\cdot \text{Weather} \\
\cdot \text{GlaiveRush} \\
\cdot \text{Critical} \\
\cdot \text{random} \\
\cdot \text{STAB} \\
\cdot \text{Type} \\
\cdot \text{Burn} \\
\cdot \text{other} \\
\cdot \text{ZMove} \\
\cdot \text{TeraShield.}
\end{aligned}
\tag{3}
$$

An example prompt that is generated by the damage calculator for all matchups is provided in listing 1.

```
1    Historical turns:
2  Battle start: You sent out Iron Crown. Opponent sent out Primarina.
3  Turn 1: Current battle state:
4  Requires switch:
5  dragapult vs. primarina:
6  dragapult outspeeds primarina
7  dragapult's moves:
8  dragondarts: 161 turns to KO opponent's pokemon
9  uturn: 6 turns to KO opponent's pokemon
10 quickattack: 5 turns to KO opponent's pokemon
11 terablast: 5 turns to KO opponent's pokemon
12 dragapult's moves if opponent's primarina uses 'terastallize':
13 dragondarts: 3 turns to KO opponent's pokemon
14 uturn: 11 turns to KO opponent's pokemon
15 quickattack: 321 turns to KO opponent's pokemon
16 terablast: 321 turns to KO opponent's pokemon
17 dragapult's moves if it uses 'terastallize' and opponent's primarina
       uses 'terastallize':
18 dragondarts: 4 turns to KO opponent's pokemon
19 uturn: 6 turns to KO opponent's pokemon
20 quickattack: 10 turns to KO opponent's pokemon
21 terablast: 9 turns to KO opponent's pokemon
22 dragapult's moves if it uses 'terastallize' and opponent's primarina
       does NOT use 'terastallize':
23 dragondarts: 161 turns to KO opponent's pokemon
24 uturn: 6 turns to KO opponent's pokemon
25 quickattack: 5 turns to KO opponent's pokemon
26 terablast: 5 turns to KO opponent's pokemon
27 Opponent moves: primarina
28 moonblast: 2 turns to KO your pokemon
```

```
29 psychicnoise: 4 turns to KO your pokemon
30 surf: 4 turns to KO your pokemon
31 flipturn: 10 turns to KO your pokemon
32
33 Requires switch:
34 primarina vs. primarina:
35 primarina outspeeds primarina
36 primarina's moves:
37 calmmind: Raises user's Special Attack and Special Defense.  3.0
       turns to KO opponent's pokemon
38 surf: 4 turns to KO opponent's pokemon
39 moonblast: 2 turns to KO opponent's pokemon
40 psychicnoise: 4 turns to KO opponent's pokemon
41 primarina's moves if opponent's primarina uses 'terastallize':
42 calmmind: Raises user's Special Attack and Special Defense.  3.0
       turns to KO opponent's pokemon
43 surf: 2 turns to KO opponent's pokemon
44 moonblast: 4 turns to KO opponent's pokemon
45 psychicnoise: 7 turns to KO opponent's pokemon
46 primarina's moves if it uses 'terastallize' and opponent's primarina
       uses 'terastallize':
47 calmmind: Raises user's Special Attack and Special Defense.  3.0
       turns to KO opponent's pokemon
48 surf: 3 turns to KO opponent's pokemon
49 moonblast: 6 turns to KO opponent's pokemon
50 psychicnoise: 7 turns to KO opponent's pokemon
51 primarina's moves if it uses 'terastallize' and opponent's primarina
       does NOT use 'terastallize':
52 calmmind: Raises user's Special Attack and Special Defense.  3.0
       turns to KO opponent's pokemon
53 surf: 6 turns to KO opponent's pokemon
54 moonblast: 3 turns to KO opponent's pokemon
55 psychicnoise: 4 turns to KO opponent's pokemon
56 Opponent moves: primarina
57 moonblast: 3 turns to KO your pokemon
58 psychicnoise: 5 turns to KO your pokemon
59 surf: 5 turns to KO your pokemon
60 flipturn: 10 turns to KO your pokemon
61
62 Current pokemon:
63 ironcrown vs. primarina:
64 ironcrown outspeeds primarina
65 ironcrown's moves:
66 futuresight: 2 turns to KO opponent's pokemon
67 tachyoncutter: 2 turns to KO opponent's pokemon
68 voltswitch: 3 turns to KO opponent's pokemon
69 focusblast: 7 turns to KO opponent's pokemon
70 ironcrown's moves if opponent's primarina uses 'terastallize':
71 futuresight: 7 turns to KO opponent's pokemon
72 tachyoncutter: 4 turns to KO opponent's pokemon
73 voltswitch: 5 turns to KO opponent's pokemon
74 focusblast: 4 turns to KO opponent's pokemon
75 ironcrown's moves if it uses 'terastallize' and opponent's primarina
       uses 'terastallize':
76 futuresight: 5 turns to KO opponent's pokemon
77 tachyoncutter: 3 turns to KO opponent's pokemon
78 voltswitch: 5 turns to KO opponent's pokemon
79 focusblast: 2 turns to KO opponent's pokemon
80 ironcrown's moves if it uses 'terastallize' and opponent's primarina
       does NOT use 'terastallize':
81 futuresight: 3 turns to KO opponent's pokemon
82 tachyoncutter: 2 turns to KO opponent's pokemon
83 voltswitch: 3 turns to KO opponent's pokemon
84 focusblast: 7 turns to KO opponent's pokemon
85 Opponent moves: primarina
```

```
86 moonblast: 7 turns to KO your pokemon
87 psychicnoise: 22 turns to KO your pokemon
88 surf: 4 turns to KO your pokemon
89 flipturn: 7 turns to KO your pokemon
90
91 Requires switch:
92 samurotthisui vs. primarina:
93 samurotthisui outspeeds primarina
94 samurotthisui's moves:
95 ceaselessedge: 5 turns to KO opponent's pokemon
96 razorshell: 5 turns to KO opponent's pokemon
97 knockoff: 5 turns to KO opponent's pokemon
98 encore: Forces opponent to keep using its last move for 3 turns.  inf
        turns to KO opponent's pokemon
99 samurotthisui's moves if opponent's primarina uses 'terastallize':
100 ceaselessedge: 5 turns to KO opponent's pokemon
101 razorshell: 3 turns to KO opponent's pokemon
102 knockoff: 5 turns to KO opponent's pokemon
103 encore: Forces opponent to keep using its last move for 3 turns.  inf
        turns to KO opponent's pokemon
104 samurotthisui's moves if it uses 'terastallize' and opponent's
        primarina uses 'terastallize':
105 ceaselessedge: 4 turns to KO opponent's pokemon
106 razorshell: 4 turns to KO opponent's pokemon
107 knockoff: 4 turns to KO opponent's pokemon
108 encore: Forces opponent to keep using its last move for 3 turns.  inf
        turns to KO opponent's pokemon
109 samurotthisui's moves if it uses 'terastallize' and opponent's
        primarina does NOT use 'terastallize':
110 ceaselessedge: 8 turns to KO opponent's pokemon
111 razorshell: 7 turns to KO opponent's pokemon
112 knockoff: 7 turns to KO opponent's pokemon
113 encore: Forces opponent to keep using its last move for 3 turns.  inf
        turns to KO opponent's pokemon
114 Opponent moves: primarina
115 moonblast: 2 turns to KO your pokemon
116 psychicnoise: 322 turns to KO your pokemon
117 surf: 4 turns to KO your pokemon
118 flipturn: 11 turns to KO your pokemon
119
120 Requires switch:
121 kingambit vs. primarina:
122 primarina outspeeds kingambit
123 kingambit's moves:
124 swordsdance: Sharply raises user's Attack.  2.0 turns to KO opponent'
        s pokemon
125 kowtowcleave: 3 turns to KO opponent's pokemon
126 suckerpunch: 4 turns to KO opponent's pokemon
127 ironhead: 2 turns to KO opponent's pokemon
128 kingambit's moves if opponent's primarina uses 'terastallize':
129 swordsdance: Sharply raises user's Attack.  2.0 turns to KO opponent'
        s pokemon
130 kowtowcleave: 2 turns to KO opponent's pokemon
131 suckerpunch: 2 turns to KO opponent's pokemon
132 ironhead: 6 turns to KO opponent's pokemon
133 kingambit's moves if it uses 'terastallize' and opponent's primarina
        uses 'terastallize':
134 swordsdance: Sharply raises user's Attack.  2.0 turns to KO opponent'
        s pokemon
135 kowtowcleave: 1 turns to KO opponent's pokemon
136 suckerpunch: 2 turns to KO opponent's pokemon
137 ironhead: 5 turns to KO opponent's pokemon
138 kingambit's moves if it uses 'terastallize' and opponent's primarina
        does NOT use 'terastallize':
```

```
139 swordsdance: Sharply raises user's Attack.  3.0 turns to KO opponent'
        s pokemon
140 kowtowcleave: 2 turns to KO opponent's pokemon
141 suckerpunch: 3 turns to KO opponent's pokemon
142 ironhead: 3 turns to KO opponent's pokemon
143 Opponent moves: primarina
144 moonblast: 3 turns to KO your pokemon
145 psychicnoise: 405 turns to KO your pokemon
146 surf: 3 turns to KO your pokemon
147 flipturn: 9 turns to KO your pokemon
148
149 Requires switch:
150 landorustherian vs. primarina:
151 landorustherian outspeeds primarina
152 landorustherian's moves:
153 stealthrock: Damages opponents switching into battle.  inf turns to
        KO opponent's pokemon
154 earthpower: 4 turns to KO opponent's pokemon
155 uturn: 6 turns to KO opponent's pokemon
156 grassknot: 4 turns to KO opponent's pokemon
157 landorustherian's moves if opponent's primarina uses 'terastallize':
158 stealthrock: Damages opponents switching into battle.  inf turns to
        KO opponent's pokemon
159 earthpower: 321 turns to KO opponent's pokemon
160 uturn: 11 turns to KO opponent's pokemon
161 grassknot: 30 turns to KO opponent's pokemon
162 landorustherian's moves if it uses 'terastallize' and opponent's
        primarina uses 'terastallize':
163 stealthrock: Damages opponents switching into battle.  inf turns to
        KO opponent's pokemon
164 earthpower: 3 turns to KO opponent's pokemon
165 uturn: 6 turns to KO opponent's pokemon
166 grassknot: 15 turns to KO opponent's pokemon
167 landorustherian's moves if it uses 'terastallize' and opponent's
        primarina does NOT use 'terastallize':
168 stealthrock: Damages opponents switching into battle.  inf turns to
        KO opponent's pokemon
169 earthpower: 5 turns to KO opponent's pokemon
170 uturn: 6 turns to KO opponent's pokemon
171 grassknot: 4 turns to KO opponent's pokemon
172 Opponent moves: primarina
173 moonblast: 3 turns to KO your pokemon
174 psychicnoise: 4 turns to KO your pokemon
175 surf: 2 turns to KO your pokemon
176 flipturn: 4 turns to KO your pokemon
177
178 'terastallize' changes a Pokemon's defensive typing to solely their
        tera type, meaning their resistances and weaknesses can change. It
         also gives them a boost to moves of their new typing. You can
        only 'terastallize' one Pokemon per battle, and it will last on
        that Pokemon until they are KO'd or the battle ends. You can
        choose to 'terastallize' and use another move in the same turn.
179 Recall the information about each of ironcrown's move actions and
        available switch actions. Which move or switch will KO the
        opponent's pokemon in the fewest turns?
180 You are able to use ['terastallize'] this turn as well.
181 It is recommended you choose to 'terastallize' this turn paired with
        a move from your available moves.
182 You have 5 pokemons:
183 [<switch_pokemon_name>] = ['landorustherian', 'dragapult', 'kingambit
        ', 'samurotthisui', 'primarina']
184  Your current Pokemon: ironcrown.
185 Choose only from the following action choices:
186 [<move_name>] = ['futuresight', 'tachyoncutter', 'voltswitch', '
        focusblast']
```

```
187 Choose the best action and your output MUST be a JSON like: {"move":"
        <move_name>"} or {"terastallize":"<move_name>"} or {"switch":"<
        switch_pokemon_name>"}
188 In fewer than 3 sentences, let's think step by step:
189
190 {"terastallize":"focusblast"}
```

Listing 1: Example prompt from damage calculator.

## D    MORE PUZZLES

### D.1    STALL STRATEGY

PokeChamp struggles with stall strategies, where opponents will try to use status effect moves to win game slowly. This is due to uncertainty in the current matchup, causing PokeChamp to choose to switch its current Pokemon in favor of another. See figure 7.

### D.2    EXCESSIVE SWITCHING

PokeChamp is struggles with capitalizing on its opponent excessively switching. Here, opponents will switch Pokémon often to prevent short lookahead methods from choosing the optimal action. See figure 7.

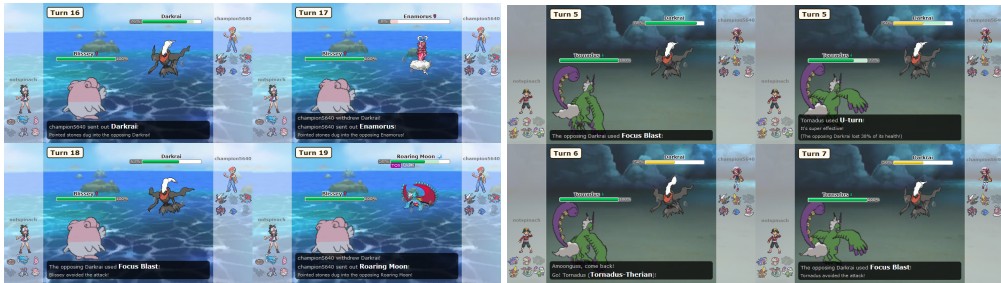

Figure 7: **Left:** *Stall strategy*: PokeChamp decides to use Darkrai against Blissey as Darkrai's Focus Blast is strong against Blissey. However, PokeChamp instead decides to switch to Enamorous, which faints from entry hazards, before sending Darkrai back in. It misses it first Focus Blast against Blissey, which causes PokeChamp to become more uncertain and decide to switch to another Pokemon. **Right:** *Opponent excessive switching strategy*: PokeChamp is unable to capitalize or defend itself from being capitalized on by a strategy where the opponent excessively switches their Pokemon. It chooses to consistently choose Focus Blast rather than switch strategies, which is exploited by switching between two Pokemon that are resistant to fighting-type attacks.

## E    RANDOM BATTLES

In this section we present experiments with PokéChamp against wbaselines in the Gen 8 Random Battles meta with and without the dynamaxing mechanic.

The results for Gen 8 Random Battles follow from the main results of the paper. Though, with less reliance on the damage calculator since the opponent-modeling from the historical data is not available. Even without this feature, PokéChamp clearly outperforms all other methods. See figure E for results with the dynamax mechanic. See figure E for results without the dynamax mechanic. See figure 8 shows the winrate of the individual method matchups.

Table 6: Gen 8 Random Battles without dynamax mechanic.

| Bot Method | Language Model | Win Rate vs. Abyssal (%) |
|---|---|---|
| **PokéChamp (Ours)** | **GPT-4o** | **70%** |
| PokéChamp (Ours) | Llama 3.1:8b | 64% |
| PokéLLMon Hu et al. (2024b) | GPT-4o | 56% |
| Dmg Calc | N/A | 44% |

Table 7: Gen 8 Random Battles with dynamax mechanic.

| Bot Method | Language Model | Win Rate vs. Abyssal (%) | Elo | Avg # Turns |
|---|---|---|---|---|
| **PokéChamp (Ours)** | **GPT-4o** | **56%** | **1273** | **17.1** |
| PokéChamp (Ours) | Llama 3.1:8b | 52% | 1184 | 19.1 |
| PokéLLMon | GPT-4o | 36% | 1048 | 22.5 |
| Abyssal | N/A | N/A | 1213 | 19.0 |
| Dmg Calc | N/A | 16% | 998 | 18.9 |
| Max Power | N/A | 4% | 787 | 23.2 |
| Random | N/A | 0% | 493 | 24.3 |

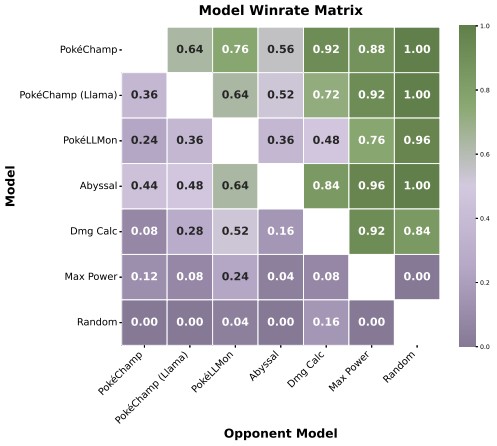

Figure 8: Gen 8 Random Battles matchup matrix per method.

