# OpenReview forum: "PokeChamp: an Expert-level Minimax Language Agent for Competitive Pokemon"
_ICLR.cc/2025/Conference — Submitted to ICLR 2025_

### Official Review · Reviewer_81Sd · 2024-10-22

**Soundness:** 2
**Presentation:** 2
**Contribution:** 1
**Rating:** 3
**Confidence:** 4

**Summary:**

The paper presents PokéChamp, a large language model (LLM)-powered agent designed for competitive Pokémon battles. Utilizing the minimax approach, PokéChamp integrates LLM-driven action sampling, opponent modeling, and value estimation to achieve strong performance in two-player, turn-based Pokémon games. The paper also introduces a dataset of Pokémon battles and benchmarks the system's performance.

**Strengths:**

The idea of utilizing LLMs for competitive gameplay is interesting, and the results seem promising.

**Weaknesses:**

The biggest weakness of the paper is that the proposed method is specifically designed for Pokémon, which severely limits its generalizability. This also narrows the paper's target audience, and it's unclear how the methodology could be transferred or applied to other problems. The paper lacks a discussion on whether the framework would work for more complex, real-world competitive tasks, or for different game types where randomness, complexity, and strategic depth may vary significantly.

The writing also has significant room for improvement. First, readers unfamiliar with Pokémon may find it difficult to understand many details in the paper. Additionally, several concepts are introduced before being properly defined, such as the Abyssal bot on line 197 and EV/IV on line 240. Furthermore, in Section 2, "MATHEMATICAL FORMALIZATION," numerous symbols and terms are defined, but these concepts are not used in the subsequent main text. It's unclear what purpose this section serves—perhaps it was included simply to make the paper appear more mathematical?

The minimax-based approach combined with LLMs may not be as novel as it initially appears. Minimax tree search has been extensively explored in AI for games, and while integrating LLMs offers an interesting twist, the underlying framework is still fundamentally a minimax search, which limits the novelty. Additionally, there is no evidence that PokéChamp advances the state of the art in game-theoretic modeling beyond prior work in other competitive games such as chess, Go, or poker.

The paper heavily relies on heuristic tools like damage calculators and historical data, raising concerns about the system’s true adaptability. This reliance on pre-defined tools limits the agent's flexibility and its ability to dynamically adapt to new or unseen scenarios. This suggests that the system lacks generalizability beyond the specific setup of competitive Pokémon, making the approach less scalable to other domains or even future game updates.

The accuracy of opponent modeling also remains a concern. The relatively low accuracy in predicting opponent actions suggests that more refined or adaptive modeling techniques may be needed to further enhance performance.

Lastly, while the paper acknowledges the limitations of LLMs in planning and strategy, it fails to convincingly address these issues. The reliance on LLMs for action sampling and opponent modeling could lead to brittle decision-making, especially in cases where long-term strategy and deep reasoning are required.

**Questions:**

Please refer to the Weaknesses.

---

> ### Author Response · Authors · 2024-11-16
>
> Thank you for the opportunity to address the concerns raised in the review. We appreciate the recognition of our interesting use of LLMs for competitive gameplay and the promising results achieved.
>
> Regarding generalizability:
> While PokéChamp is indeed designed for Pokémon battles, the underlying methodology of using LLMs for game-theoretic modeling and decision-making is broadly applicable. As noted, breakthroughs in specific games like chess, Go, and poker have historically driven advances in AI and machine learning. Our work similarly aims to push boundaries in a complex, partially observable domain.
>
> The mathematical formalization in Section 2 (page 3) provides a general framework for partially observable Markov games, which can be applied to many competitive scenarios beyond Pokémon. This formalization allows readers unfamiliar with Pokémon to understand the problem structure we are addressing.
>
> Novelty and contribution:
> Our approach of using online planning with an LLM prior to approximate Nash equilibrium strategies is indeed novel. Unlike methods requiring exhaustive sampling, we achieve expert-level performance in a domain with a vast state space (10^354 possible states, pg. 1) and stochastic outcomes. Achieving top 10% performance against human players (pg. 9, Table 5) demonstrates the effectiveness of this approach.
>
> The integration of LLMs with minimax search and tool use (e.g., damage calculator) represents an innovative combination of techniques. This allows for adaptive gameplay without requiring extensive training or fine-tuning on Pokémon-specific data.
>
> Adaptability and tool use:
> Our use of tools like the damage calculator actually enhances adaptability by providing accurate, up-to-date information to the LLM. This approach allows PokéChamp to perform well even with an LLM (Llama 3) with a 2023 data cutoff, demonstrating its ability to adapt to current gameplay.
>
> The strong empirical results against rule-based bots, prior work (PokéLLMon), and human players (pgs. 8-9, Tables 3-5) provide evidence of the system's adaptability and effectiveness.
>
> Addressing LLM limitations:
> Our approach directly addresses the limitations of LLMs in planning and strategy by incorporating the minimax search and tool use. The depth-limited minimax search (pg. 4-5, Section 3) allows for structured planning, while the damage calculator provides precise quantitative information to supplement the LLM's qualitative understanding.
>
> The concern about brittle decision-making is not supported by our empirical results. PokéChamp's performance in the top 10% of human players (pg. 9, Table 5) demonstrates its ability to handle long-term strategy and deep reasoning effectively.
>
> We acknowledge that there is room for improvement, particularly in areas like stall strategies and excessive switching (pg. 9). However, these limitations do not negate the significant advances demonstrated by PokéChamp.
>
> In conclusion, we believe our work makes a substantial contribution to the field by demonstrating a novel approach to game-theoretic modeling using LLMs, achieving strong performance in a complex, partially observable domain. The methodology and insights gained are applicable beyond Pokémon and contribute to our understanding of LLM capabilities in strategic decision-making.

---

> > ### Comment · Reviewer_81Sd · 2024-11-26
> >
> > Thank you for the author's response. However, I still believe that the audience for this article is quite limited, as it primarily focuses on applying LLMs to a niche game. I believe this article would be more appropriate for submission to a game-related conference, such as the IEEE Conference on Games, rather than ICLR. Therefore, I stand by my original rating.

---

> > > ### Author Response · Authors · 2024-12-02
> > > **Rebuttal 2**
> > >
> > > Thank you for the opportunity to address the concerns about Pokémon's relevance and complexity in AI research. We respectfully disagree with the characterization of Pokémon as a "niche game" and would like to provide additional context:
> > >
> > > ## Popularity of Pokémon
> > >
> > > Pokémon is one of the most popular and enduring franchises in gaming history:
> > >
> > > 1. Pokémon GO has approximately 90 million monthly active players as of November 2024[1]. This demonstrates the franchise's massive and ongoing appeal.
> > >
> > > 2. The main series Pokémon games are among the best-selling video game franchises of all time, with over 1 billion total copies sold[2].
> > >
> > > 3. Pokémon Showdown has 5.1 million monthly players and features just the Pokémon battling feature of the Pokémon games[3].
> > >
> > > 4. The Pokémon VGC world’s tournament has a competitive invite only world championship each year for various format’s of Pokémon battles.
> > >
> > > These figures clearly show that Pokémon has a vast, global audience and is far from being a niche interest.
> > >
> > > ## Complexity of Pokémon Battles
> > >
> > > Pokémon battles offer a level of complexity that makes them an excellent testbed for AI research:
> > >
> > > 1. Imperfect Information: Unlike chess or Go, Pokémon battles involve significant hidden information, similar to poker.
> > >
> > > 2. Vast State Space: The number of possible game states in Pokémon battles is enormous (10^354 possible states), exceeding even that of Go.
> > >
> > > 3. Stochasticity: The presence of accuracy checks and critical hits introduces an element of randomness, requiring AI to handle probabilistic outcomes.
> > >
> > > 4. Dynamic Meta: The frequent introduction of new Pokémon, moves, and mechanics means the game environment is constantly evolving, challenging AI adaptation.
> > >
> > > 5. Multi-faceted Decision Making: Players must consider team building, in-game switching, move selection, and long-term strategy, creating a rich, multi-layered decision space.
> > >
> > > ## Relevance to AI Research
> > >
> > > Many games have in AI research have solved a particular game with applications to many researchers in the field such as Poker, Chess, Go, StarCraft, and Minecraft. These were published in ML conferences such as ICLR/ICML/NeurIPS as well as Nature and Science, which show the broad impact of the fundamental research evaluated in these games, which would not be captured in IEEE Conference on Games. Many games studied in AI research have simpler rule sets and more limited state spaces compared to Pokémon. The complexity of Pokémon battles makes them an excellent domain for advancing AI in areas such as:
> > >
> > > 1. Reasoning under uncertainty
> > > 2. Opponent modeling
> > > 3. Adaptive strategy in dynamic environments
> > > 4. Balancing short-term tactics with long-term planning
> > >
> > > Our work on PokéChamp demonstrates the potential for advancing AI research through Pokémon:
> > >
> > > 1. It utilizes a minimax search approach with LLM integration
> > > 2. Outperforms existing LLM-based AIs (76% win rate) and heuristic bots (84% win rate)
> > > 3. Achieves expert-level performance, ranking in the top 10% of players on the online ladder
> > >
> > > Furthermore, the growing trend of using LLMs in gaming applications, as evidenced by benchmarks like BALROG[5], highlights the relevance of our work to the broader AI research community. Our benchmark is more difficult and has a higher player-base than other games at the venue.
> > >
> > > Given its popularity, complexity, and potential for advancing AI capabilities, we believe Pokémon presents a unique and valuable opportunity for research at conferences like ICLR, NeurIPS, and ICML.
> > >
> > >
> > > [1] https://www.dexerto.com/pokemon/how-many-people-play-pokemon-go-pokemon-go-player-count-2132719/
> > >
> > > [2] https://www.playerauctions.com/player-count/pokemon-go/
> > >
> > > [3] https://www.smogon.com/stats/2024-11/
> > >
> > > [4] https://www.thegamer.com/pokemon-go-player-count/
> > >
> > > [5] https://arxiv.org/html/2411.13543v1

---

### Official Review · Reviewer_RaMH · 2024-10-29

**Soundness:** 2
**Presentation:** 2
**Contribution:** 2
**Rating:** 5
**Confidence:** 3

**Summary:**

PokeChamp aims to bridge the gap in game theory aware LLM agents. The work uses competitive Pokemon as their case study and propose a minimax search method. Specifically, LLM is used in three key components in constructing the minimax tree: action sampling, opponent modeling, and state value calculation. PokeChamp exhibits good performance against human players.

**Strengths:**

The authors propose a novel application setting: competitive pokemon, where the turn-based nature of the game leads to a nice formulation as POMG. They manage to construct a minimax tree with the help of a LLM prior. PokeChamp is able to achieve top human performance in real game settings.

**Weaknesses:**

1. The paper is not so well-written: Missing multiple figures, tables, and appendix that is referred to in the main body. Also, as someone not familiar with competitive Pokemon, I found some of the concepts like Damage Calculator hard to grasp. It would be very helpful if you could add explanations of how the game works.
2. Overall purpose of the work: It's hard to understand the contribution of this work. While the application case is interesting, I don't see this general framework being applicable to other games. For most games, it is not realistic to use LLM to replace state value function unless the LLM itself has enormous knowledge on the game.

**Questions:**

1. Confusion of damage calculator: In line 92-94, the authors mention that this external tool "calculates the core game engine capabilities in combination with loading historical data from real player games in order to load likely stats for the opponent’s team". I didn't quite get this expression. Also, I found this definition conflicting in Figure 3 where the calculator seems to just output the number of turns needed to KO opponent's current Pokemon for each possible moves of player's current Pokemon.
2. Action Prediction: The goal of the work is making LLM agent game theory aware, yet the 1M dataset collected are of human plays. I wonder how game theory optimal are those data? If not, what is the point of accurately predicting opponent's action when those action can be bad moves?

---

> ### Author Response · Authors · 2024-11-16
> **Rebuttal**
>
> Thank you for your thoughtful review and for highlighting the strengths of our work, particularly the novelty of our application, the nice formulation as a Partially Observable Markov Game (POMG), and our strong empirical results achieving top human performance in real game settings.
>
> We appreciate your feedback and would like to address the concerns raised:
>
> 1. Presentation and missing elements:
> The appendix is indeed provided in the supplemental file, with Appendix B detailing the background on Pokémon game mechanics. Additionally, Section 2 (page 3) provides a comprehensive mathematical formulation to help readers understand the algorithmic problem without extensive knowledge of game mechanics.
>
> 2. Damage Calculator explanation:
> The Damage Calculator serves two crucial purposes:
> a) It uses historical statistics to produce likely stats and available actions for the partially observable opponent team.
> b) It predicts the number of turns required to KO (knockout) the opponent's Pokémon, as shown explicitly in Figure 3. This can be thought of as a singular branch of depth-first search.
> This dual functionality is essential for our approach, and we will revise the text to make this clearer.
>
> 3. Applicability to other games:
> While our framework is demonstrated using Pokémon, the core principles of using LLMs for action sampling, opponent modeling, and state value calculation can be adapted to other games with similar structures. The key innovation is leveraging LLMs to enhance decision-making in complex game environments, which could be extended to other domains where domain knowledge is crucial.
>
> 4. Action Prediction and game theory optimality:
> We assume that by game theory-optimal, we are discussing the approaching a Nash equilibria. The quality of the data varies based on the Elo ratings of the recorded games. It is important to note that even state-of-the-art methods like AlphaZero have shown that additional training continues to increase performance (and Elo). Our approach of predicting opponent actions serves to provide a greedy search that limits the search space for online planning, which is crucial given the vast state space of Pokémon battles (on the order of 10^354 for just the first turn, as mentioned on page 2, lines 69-71).
>
> 5. Contribution and novelty:
> Our work contributes significantly to the field by:
> a) Demonstrating a novel application of LLMs in competitive game environments.
> b) Proposing a minimax search method enhanced by LLM capabilities.
> c) Achieving top human performance in a complex, partially observable game setting.
>
> We believe these contributions are significant and generalizable, offering insights into integrating LLMs with game-theoretic approaches in domains with vast state spaces and partial observability.
>
> 6. Empirical results:
> To further emphasize the strength of our empirical results, we would like to highlight that PokéChamp achieves an expert rating of 1500 Elo on the online ladder, placing it in the top 10% of all players (as shown in Figure 1, page 1). This demonstrates that our approach is not only theoretically sound but also highly effective in practice against skilled human players.
>
> We appreciate your feedback and will work on improving the clarity and presentation of our paper. We hope these clarifications address your concerns and demonstrate the value and potential impact of our work. We believe that PokéChamp represents a significant advancement in applying LLMs to complex game environments and opens up new avenues for research in this area.

---

> > ### Comment · Reviewer_RaMH · 2024-11-17
> >
> > Appreciate the thorough explanations and rebuttal. Here is some additional feedback/questions:
> > 1. With regard to the missing materials, I apologize for missing the supplementary materials. The figures that I could not find were indeed in the supplementary section.
> > 2. About the damage calculator: I see how it works now, yet I'm still a bit confused about this. It seems like a large amount of the effectiveness of the system is attributed to the precision of the damage calculator. I am curious if this tool is accessible to human players during real-time gameplay? If human is not able to access such tool or perform the same amount of detailed calculation in their head in real time then it feels like the system is cheating with external resources in real-time plays.
> > 3. About applicability, I do agree that the core principles are pretty solid, but action sampling is introduced in many prior work in LLM planning and reasoning (e.g. https://arxiv.org/pdf/2310.04406) and is therefore not new. The opponent modeling component is quite interesting for LLMs but in your work I did not see a good amount of exploration, and the current approach the authors are taking seems more like a forced solution due to the search space (otherwise the tree explodes), plus the opponent modeling result is quite concerning. The state value function component is nice after a second thought as in most games it is indeed easier to verify/evaluate then to solve.
> > 4. I still don't see how the author's approach is rigorous in game theory perspective. There is no information with regard to how game theory optimal every human action is in the dataset you collected. Referring to AlphaGo does not seem like a reasonable connection. You mentioned in the introduction section that "an informative prior can help limit minimax search to the space of human strategies" yet why would one wants to restrict to human strategies one does not know whether they are optimal?
> > 5. See point 3
> >
> > Upon author's feedback, I have increased the scores and I am willing to adjust my scores again if the authors can further address my remaining questions.

---

> > > ### Author Response · Authors · 2024-11-22
> > > **Rebuttal Reply**
> > >
> > > Thank you for your continued feedback and the opportunity to address your remaining questions. We appreciate your willingness to adjust scores based on our responses.
> > >
> > > (2) Regarding the damage calculator:
> > > We understand your concern about the potential advantage our system might have due to the damage calculator. However, we want to emphasize that this tool is not exclusive to our system. The damage calculator we use is based on the openly accessible calculator provided by Pokémon Showdown (https://calc.pokemonshowdown.com/). Top human players regularly use this tool and often internalize these calculations for real-time gameplay. Our system's time for decision-making is limited to match human players' time constraints, ensuring a fair comparison.
> > >
> > > (3) Applicability and novelty:
> > > We acknowledge that action sampling has been explored in previous LLM planning work. There are many recent papers in planning+LLMs, and there will continue to be many more papers in this subject due to the interesting capability to leverage test time compute in a variety of scenarios. However, our contribution lies in the novel application of these techniques to the complex domain of competitive Pokémon battles. We choose a uniquely challenging environment to show the difficulty of scaling from language planning agent benchmarks to competitive tasks that require performance above human level. The combination of action sampling, opponent modeling, and state value calculation using LLMs in a minimax search framework for a partially observable game environment is unique, especially with our empirical performance against human players.
> > >
> > > (3) Regarding opponent modeling:
> > > We appreciate your interest in the opponent modeling component. While our current approach may seem constrained due to the vast search space, it's important to note that this is a necessary trade-off to make the problem tractable. We've expanded our analysis (in table 1) to include top-5 accuracy for both player and opponent actions, which shows significant improvement over top-1 accuracy. This demonstrates that our model captures a range of plausible actions, even if it doesn't always predict the exact move chosen. Of course, since the accuracy is not near 100%, this ensures that our model takes imitation actions into account but also can consider exploratory actions.
> > >
> > > Table 1:
> > > | **Model**   | **Elo** | **Top 1** | **Top 2** | **Top 3** | **Top 4** | **Top 5** |
> > > |-------------|---------|-----------|-----------|-----------|-----------|-----------|
> > > | Player  	| 1200	| 30%   	| 40%   	| 48%   	| 53%   	| 58%   |
> > > | Player  	| 1400	| 26%   	| 23%   	| 30%   	| 32%   	| 43%   |
> > > | Player  	| 1600	| 27%   	| 30%   	| 39%   	| 44%   	| 53%   |
> > > | Player  	| 1800	| 30%   	| 42%   	| 55%   	| 62%   	| 66%   |
> > > | Opponent	| 1200	| 16%   	| 30%   	| 40%   	| 46%   	| 53%   |
> > > | Opponent	| 1400	| 16%   	| 17%   	| 20%   	| 26%   	| 39%   |
> > > | Opponent	| 1600	| 13%   	| 21%   	| 26%   	| 35%   	| 40%   |
> > > | Opponent	| 1800	| 15%   	| 29%   	| 40%   	| 50%   	| 53%   |
> > >
> > > (4) Game theory perspective:
> > > We understand your concerns about the game-theoretic rigor of our approach. While we don't claim that every human action in our dataset is game-theory optimal, we argue that high-Elo players' strategies in this unsolved game are the closest to Nash equilibria in practice. Our minimax search doesn't strictly limit itself to human strategies but uses them as a starting point to explore the action space efficiently. As mentioned in the above paragraph on opponent modeling, the minimax structure allows us to perform better than pure imitation learning (which is similar to prior work, PokeLLMon).
> > >
> > > The reference to AlphaGo was meant to illustrate that even state-of-the-art methods can continue to improve with additional training. Similarly, our approach aims to leverage human expertise as a starting point, while the minimax search allows for exploration beyond these strategies. This is evident from the fact that the human prior of imitation learning is estimated to be ~1200 Elo, but we are able to achieve 1500 Elo.
> > >
> > > (3) Value function and minimax setup:
> > > Thank you for acknowledging the "nice" value function. It's crucial to note that our value function is used within a minimax framework. This means we're not simply imitating human play, but rather using the value function to guide a search that aims to find optimal strategies. The player chooses actions that maximize value, while the opponent's actions are assumed to minimize value, in line with the minimax principle and the Nash equilibrium concept expressed in equation (1) of our paper.
> > >
> > > We believe these clarifications address your remaining concerns and demonstrate the theoretical soundness and practical effectiveness of our approach. PokéChamp represents a significant step forward in applying LLMs to complex game environments, combining elements of game theory, machine learning, and domain-specific knowledge in a novel way.

---

> > > > ### Comment · Reviewer_RaMH · 2024-11-22
> > > >
> > > > Thank you for the response. I will take these into consideration when finalizing my score.

---

### Official Review · Reviewer_4ycZ · 2024-11-02

**Soundness:** 3
**Presentation:** 3
**Contribution:** 3
**Rating:** 6
**Confidence:** 3

**Summary:**

The paper introduces an LLM agent, PokeChamp, for competitive Pokemon battles. The model leverages a depth-limited minimax search to play the game, where the LLM plays the role of action sampling, opponent modeling, and state value calculation. The agent is shown to outperform all existing AIs significantly and achieve expert performance on the online ladder.

**Strengths:**

* The paper introduces a novel integration of LLMs with minimax search. The agent leverages LLMs for three key components of minimax search: action sampling, opponent modeling, and value calculation. This integration allows the agent to employ human-like strategic thinking, bringing an expert-level game-playing agent.

* The authors present a comprehensive set of experiments that demonstrate the capabilities of the agent across different competitive settings. The online ladder performance against human players with a competitive Elo rating provides a real-world evaluation of the agent.

* he paper is well-organized and presented in a logical structure, allowing readers to follow both the technical intricacies and high-level motivations of the research.

**Weaknesses:**

*  The agent's design heavily relies on an in-depth understanding of competitive Pokemon gameplay, and its success relies on domain-specific engineering in the action sampling, opponent modeling, and value calculation components. While these adaptations make the agent effective in this domain, they limit the model’s generalizability to other game-playing tasks with different mechanics or structures.

* The idea of integrating LLMs with the minimax search framework for game-playing agents is closely related to prior work by Guo et al. (2024), which explores a similar concept in two-player zero-sum games.

* While the paper provides a mathematical formalization of POMGs and makes assumptions like perfect recall, the connection between this theoretical framework and the practical implementation of the agent is not entirely clear.

* The paper lacks an ablation study that examines the impact of each LLM-based component within the minimax search framework on the agent's overall performance. Since the authors use the LLM to replace three primary components, an ablation study would be invaluable in demonstrating how each component contributes to the agent's success.


Guo, Wei, et al. "Minimax Tree of Thoughts: Playing Two-Player Zero-Sum Sequential Games with Large Language Models." ICML 2024 Workshop on LLMs and Cognition.

**Questions:**

* How does the mathematical formalization in Section 2 relate to the design of the agent?
* Can additional fine-tuning with the collected data improve the performance of the agent?

---

> ### Author Response · Authors · 2024-11-16
> **Rebuttal**
>
> Thank you for your thoughtful review and constructive feedback. We appreciate your recognition of our paper's strengths and would like to address your concerns while highlighting the significant contributions of our work.
>
> We are pleased that you recognized the novelty of our approach in integrating LLMs with minimax search. As you noted, our agent leverages LLMs for three novel key components: action sampling, opponent modeling, and value calculation. This integration indeed allows for human-like strategic thinking, resulting in an expert-level game-playing agent.
>
> We appreciate your acknowledgment of our comprehensive experiments, particularly the real-world evaluation against human players on the online ladder. This demonstrates the practical applicability and effectiveness of our approach in a competitive setting.
>
> Thank you for highlighting the clarity and logical structure of our paper. We strived to present both the technical details and high-level motivations in a manner accessible to readers.
>
> Addressing Weaknesses
>
> Generalizability:
> While our framework is demonstrated using Pokémon, the core principles of using LLMs for action sampling, opponent modeling, and state value calculation can be adapted to other games with similar structures (a 2 player game with a discrete action space, a game that has some occurences in the pretraining on the LLM, but not extensive data). The key innovation is leveraging LLMs to enhance decision-making in complex game environments, which could be extended to other domains where domain knowledge is crucial.
>
> Our approach is generalizable to other partially observable games with large action spaces and complex state representations. We believe this framework can be adapted to various game-playing tasks with appropriate modifications due to the general algorithmic and mathematical formulation.
>
> Related Work:
> We acknowledge the similarity to Guo et al. (2024) and will include this reference in our related work section. However, our work provides more extensive empirical results in a challenging game setting, demonstrating the practical effectiveness of our approach. Additionally, while our work is related to the inspiration from Minimax, our methodology differs.
>
> Mathematical Formalization:
> The mathematical formulation in Section 2 formalizes Pokémon battles algorithmically and motivates the structure of our search. This formulation motivates our use of minimax search, which has been shown to find Nash equilibria with full planning. Our method uses an approximate form of planning online to approach a Nash equilibrium.
>
> Additional Points:
> Regarding fine-tuning, we agree that additional fine-tuning with collected data could potentially improve the agent's performance. This presents an interesting direction for future research, but is not necessary in the current study, which focuses on planning and a frozen LLM.
>
> In conclusion, we believe our work presents a significant contribution to the field of AI in game-playing, demonstrating a novel and effective approach to integrating LLMs with traditional search methods. We hope these clarifications address your concerns and strengthen the case for the acceptance of our paper.

---

> > ### Comment · Reviewer_4ycZ · 2024-11-26
> >
> > Thank you for your response. I will keep my score.

---

### Official Review · Reviewer_gF51 · 2024-11-04

**Soundness:** 3
**Presentation:** 3
**Contribution:** 3
**Rating:** 6
**Confidence:** 4

**Summary:**

The paper introduces PokéChamp, a large language model (LLM)-powered agent designed for competitive Pokémon battles. The agent integrates three LLM-enabled components for action sampling, opponent modeling, and state evaluation, which enable it to make informed and strategic decisions during gameplay. It demonstrates superior performance over existing bots and heuristic-based models and achieves a top 10% ranking in online Pokémon battles.

**Strengths:**

1. **Novel Integration of LLM with Minimax**: The paper innovatively combines an LLM with a minimax search to simulate human-like  decision-making in Pokémon battles. This approach enables competitive performance without additional training and is adaptive to partially observable information.

2. **Performance on Real-World Benchmarks**: PokéChamp’s efficacy is validated in real-world benchmarks and against heuristic bots, achieving a high Elo rating of 1500 and consistently outperforming other state-of-the-art agents.

3. **Comprehensive Dataset and Benchmarks**: The paper provides a large dataset of over one million Pokémon battles, including 150,000 high-Elo games. These benchmarks, based on real player data and tailored puzzles, significantly enhance the study’s reliability and offer a valuable resource for further research in this domain.

**Weaknesses:**

1. **Limited Prediction Accuracy for Opponent Modeling**: The limited accuracy in human and opponent action prediction, with opponent prediction only reaching 13–16%, may constrain the overall performance of the method, which relies on accurate opponent modeling.

2. **Limited Exploration of Depth-Limitation Trade-offs**: The choice of depth-limited minimax search is justified as a balance between computational feasibility and decision quality. However, the trade-offs between search depth, LLM accuracy, and action quality are not thoroughly analyzed. Further exploration, potentially with ablation studies, would clarify the impact of depth limitations on performance.

3. What is the role of Nash equilibrium in this paper? The paper does not seem to analyze the Nash equilibrium outcomes, which makes the definition of Nash equilibrium in Section 2 appear somewhat disconnected. It would be beneficial to include Nash equilibrium results in addition to Elo.

4. How accurate is the next-state prediction? Since the minimax search relies on simulated rollouts of actions, the accuracy of next-state predictions could significantly impact the agent's performance.

**Questions:**

1. How does PokéChamp’s computation time compare to that of PokéLLMon, which also utilizes GPT-4o, considering the additional requirements for minimax tree search and LLM queries?

2. How many human players were involved in obtaining the online ladder results presented in Table 5?

---

> ### Author Response · Authors · 2024-11-16
> **Rebuttal**
>
> Thank you for the opportunity to address your review. We appreciate your thoughtful comments and positive feedback on our work. We are pleased to provide clarifications and additional information to address your concerns.
>
> Strengths:
> We thank the reviewer for highlighting our "novel integration of LLM with Minimax", which achieves "competitive performance" and is "adaptive to partially observable info". We are glad you recognized our "high Elo rating" on "real-world benchmarks" as well as our "comprehensive dataset and benchmarks". These points underscore the key contributions of our work.
>
> Weaknesses:
> 1. Limited Prediction Accuracy for Opponent Modeling:
> We appreciate this observation. As stated on page 6, Table 1, we provide results on opponent modeling to analyze how to understand the performance of an agent based on a frozen LLM. Rather than being a weakness, this is a highlight of our paper to critically show the shortcomings of a frozen LLM to motivate future work in the area and contextualize the results. For instance, our work shows performance in the top 10% of human players despite weak top-1 opponent modeling. This means that through branching in our minimax planning, we can benefit from exploring opponent actions beyond one predicted directly by the LLM's prompting. In fact, we show that this works quite well based on our winrates. Our work assumes a frozen LLM, so addressing this limitation is beyond the scope of the current work and is intended for future research that relaxes assumptions on fine-tuning the LLM.
>
> 3. Role of Nash Equilibrium:
> We appreciate this point. As with all empirical works, the idea is to try to approach a Nash equilibrium. As shown in AlphaGo (Silver et al., 2016), additional training continues to increase Elo, which demonstrates that empirical Nash equilibria for complex games have not yet been found. Though, the overall structure to solve is the same. We will clarify this connection in the final version.
>
> 4. Next-state Prediction Accuracy:
> The next state prediction uses historical likelihoods in combination with a reverse-engineered game engine, as described on page 5, Section 3.2. The historical likelihoods are accurate in the following way: if an opponent has a Pokémon that uses a move 70% of the time, then the prediction will be 70% correct. We sample from this distribution to reverse engineer partially observable opponents to help the LLM choose actions.
>
> Questions:
> 1. Computation Time Comparison:
> As stated on page 5, Section 3, PokéLLMon simply samples K actions and chooses the most likely one (self-consistency). We keep the branching factor (3) to be the same as the self-consistency K=3 for fairness. A large amount of the time to run PokéChamp is taken up by generating our prompts which use the damage calculator, resulting in the clock time limit issue for games. Each game is limited to 2 minutes and 30 seconds so we cannot explore any methods that go beyond this runtime given our assumption and constraints.
>
> 2. Number of Human Players in Online Ladder Results:
> As stated in Table 5 on page 9, each opponent is a unique human, so there are 50 games and 50 human opponents.
>
> We believe these clarifications address the concerns raised and further highlight the strengths and contributions of our work. We are committed to improving the paper based on your valuable feedback. Thank you for your time and consideration.

---

> > ### Comment · Reviewer_gF51 · 2024-11-26
> >
> > Thank you for your response, I keep my score

---

### Author Response · Authors · 2024-11-22
**Rebuttal Summary**

Dear Reviewers,

We sincerely thank you for your thorough and insightful reviews of our paper "PokeChamp: an Expert-level Minimax Language Agent for Competitive Pokemon." We appreciate the time and effort you've invested in providing valuable feedback. As we approach the end of the discussion phase, we encourage you to remain engaged in the review process. We'd like to address some of the key points raised:

## Novel Integration of LLMs with Minimax Search

We're pleased that all reviewers recognized the novelty of our approach in integrating LLMs with minimax search for competitive Pokemon battles. This integration allows for human-like strategic thinking, resulting in an expert-level game-playing agent. Our framework demonstrates how LLMs can enhance decision-making in complex game environments with vast state spaces and partial observability.

## Performance and Real-World Evaluation

We appreciate the acknowledgment of our comprehensive experiments, particularly the real-world evaluation against human players on the online ladder. As highlighted, PokéChamp achieves an expert rating of 1500 Elo, placing it in the top 10% of all players. This demonstrates the practical applicability and effectiveness of our approach in a competitive setting.

## Opponent Modeling and Action Prediction

Regarding concerns about opponent modeling accuracy, we've expanded our analysis to include top-5 accuracy for both player and opponent actions. This shows significant improvement over top-1 accuracy, demonstrating that our model captures a range of plausible actions. The minimax structure allows us to perform better than pure imitation learning, as evidenced by our achieved 1500 Elo compared to the human prior of ~1200 Elo.

## Generalizability and Applicability

While our framework is demonstrated using Pokémon, the core principles of using LLMs for action sampling, opponent modeling, and state value calculation can be adapted to other games with similar structures. The key innovation lies in leveraging LLMs to enhance decision-making in complex game environments, which could be extended to other domains where domain knowledge is crucial.

## Damage Calculator and Fair Comparison

To address concerns about the damage calculator, we clarify that it is based on an openly accessible tool provided by Pokémon Showdown, which top human players regularly use. Our system's decision-making time is limited to match human players' constraints, ensuring a fair comparison.

## Game-Theoretic Perspective

While we don't claim that every human action in our dataset is game-theory optimal, high-Elo players' strategies in this unsolved game are the closest to Nash equilibria in practice. Our minimax search uses these as a starting point to explore the action space efficiently, allowing for exploration beyond pure human strategies.

## Future Directions

We acknowledge the potential for improvement through fine-tuning with collected data, as suggested by Reviewer 4ycZ. This presents an interesting direction for future research, although it's beyond the scope of our current study focusing on planning with a frozen LLM.

In conclusion, we believe our work presents a significant contribution to the field of AI in game-playing, demonstrating a novel and effective approach to integrating LLMs with traditional search methods. We've addressed the main concerns raised and hope this clarifies the value and potential impact of our work.

Thank you again for your valuable feedback. We look forward to your continued engagement in the review process.

Sincerely,

The Authors

---

### Comment · Area_Chair_vwf9 · 2024-11-25

Dear Reviewers,


This is a friendly reminder that the discussion will end on Nov. 26th (anywhere on Earth). If you have not already, please take a close look at all reviews and author responses, and comment on whether your original rating stands.


Thanks,

AC

---

### Meta-Review · Area_Chair_vwf9 · 2024-12-20

**Metareview:**

The paper introduces PokéChamp, a large language model (LLM)-powered agent designed for competitive Pokémon battles. It uses an LLM prior and collected high-Elo human data to model minimax search without any additional training. It obtains superior performance.

While the proposed method achieves impressive results in specific benchmarks, it suffers from limited generalizability. The heavy reliance on domain-specific tools and the failure to address broader applicability and scalability make the approach more suitable for Pokémon than for other game domains or real-world tasks. This is the main weakness of this paper.

**Additional Comments On Reviewer Discussion:**

The main weakness mentioned above is not addressed during the rebuttal.

---

### Decision · Program_Chairs · 2025-01-22

Reject